# Deep Visual Analogy-Making

**Scott Reed**   **Yi Zhang**   **Yuting Zhang**   **Honglak Lee**
University of Michigan, Ann Arbor, MI 48109, USA
{reedscot,yeezhang,yutingzh,honglak}@umich.edu

## Abstract

In addition to identifying the content within a single image, relating images and generating related images are critical tasks for image understanding. Recently, deep convolutional networks have yielded breakthroughs in predicting image labels, annotations and captions, but have only just begun to be used for generating high-quality images. In this paper we develop a novel deep network trained end-to-end to perform visual analogy making, which is the task of transforming a query image according to an example pair of related images. Solving this problem requires both accurately recognizing a visual relationship and generating a transformed query image accordingly. Inspired by recent advances in language modeling, we propose to solve visual analogies by learning to map images to a neural embedding in which analogical reasoning is simple, such as by vector subtraction and addition. In experiments, our model effectively models visual analogies on several datasets: 2D shapes, animated video game sprites, and 3D car models.

## 1   Introduction

Humans are good at considering "what-if?" questions about objects in their environment. What if this chair were rotated 30 degrees clockwise? What if I dyed my hair blue? We can easily imagine roughly how objects would look according to various hypothetical questions. However, current generative models of images struggle to perform this kind of task without encoding significant prior knowledge about the environment and restricting the allowed transformations.

Often, these visual hypothetical questions can be effectively answered by analogical reasoning.[1] Having observed many similar objects rotating, one could learn to mentally rotate new objects. Having observed objects with different colors (or textures), one could learn to mentally re-color (or re-texture) new objects.

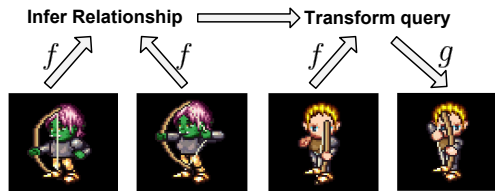

Figure 1: Visual analogy making concept. We learn an encoder function $f$ mapping images into a space in which analogies can be performed, and a decoder $g$ mapping back to the image space.

Solving the analogy problem requires the ability to identify relationships among images and transform query images accordingly. In this paper, we propose to solve the problem by directly training on visual analogy completion; that is, to generate the transformed image output. Note that we do not make any claim about how humans solve the problem, but we show that in many cases thinking by analogy is enough to solve it, without exhaustively encoding first principles into a complex model.

We denote a valid analogy as a 4-tuple A : B :: C : D, often spoken as "A is to B as C is to D". Given such an analogy, there are several questions one might ask:

- $A ? B :: C ? D$ - What is the common relationship?

- $A : B ? C : D$ - Are A and B related in the same way that C and D are related?

- $A : B :: C : ?$ - What is the result of applying the transformation $A : B$ to $C$?

The first two questions can be viewed as discriminative tasks, and could be formulated as classification problems. The third question requires generating an appropriate image to make a valid analogy. Since a model with this capability would be of practical interest, we focus on this question.

Our proposed approach is to learn a deep encoder function $f : \mathbb{R}^D \to \mathbb{R}^K$ that maps images to an embedding space suitable for reasoning about analogies, and a deep decoder function $g : \mathbb{R}^K \to \mathbb{R}^D$ that maps from the embedding back to the image space. (See Figure 1.) Our encoder function is inspired by word2vec [21], GloVe [22] and other embedding methods that map inputs to a space supporting analogies by vector addition. In those models, analogies could be performed via

$$d = \arg\max_{w \in \mathcal{V}} \cos(f(w), f(b) - f(a) + f(c))$$

where $\mathcal{V}$ is the vocabulary and $(a, b, c, d)$ form an analogy tuple such that $a : b :: c : d$. Other variations, such as a multiplicative version [18], on this inference have been proposed. The vector $f(b) - f(a)$ represents the transformation, which is applied to a query $c$ by vector addition in the embedding space. In the case of images, we can modify this naturally by replacing the cosine similarity and $argmax$ over the vocabulary with application of a decoder function mapping from the embedding back to the image space.

Clearly, this simple vector addition will not accurately model transformations for low-level representations such as raw pixels, and so in this work we seek to learn a high-level representation. In our experiments, we parametrize the encoder $f$ and decoder $g$ as deep convolutional neural networks (CNN), but in principle other methods could be used to model $f$ and $g$. In addition to vector addition, we also propose more powerful methods of applying the inferred transformations to new images, such as higher-order multiplicative interactions and multi-layer additive interactions.

We first demonstrate visual analogy making on a 2D shapes benchmark, with variation in shape, color, rotation, scaling and position, and evaluate the performance on analogy completion. Second, we generate a dataset of animated 2D video game character sprites using graphics assets from the Liberated Pixel Cup [1]. We demonstrate the capability of our model to transfer animations onto novel characters from a single frame, and to perform analogies that traverse the manifold induced by an animation. Third, we apply our model to the task of analogy making on 3D car models, and show that our model can perform 3D pose transfer and rotation by analogy.

## 2 Related Work

Hertzmann et al. [12] developed a method for applying new textures to images by analogy. This problem is of practical interest, e.g., for stylizing animations [3]. Our model can also synthesize new images by analogy to examples, but we study global transformations rather than only changing the texture of the image.

Dollár et al. [9] developed Locally-Smooth Manifold Learning to traverse image manifolds. We share a similar motivation when analogical reasoning requires walking along a manifold (e.g. pose analogies), but our model leverages a deep encoder and decoder trainable by backprop.

Memisevic and Hinton [19] proposed the Factored Gated Boltzmann Machine for learning to represent transformations between pairs of images. This and related models [25, 8, 20] use 3-way tensors or their factorization to infer translations, rotations and other transformations from a pair of images, and apply the same transformation to a new image. In this work, we share a similar goal, but we directly train a deep predictive model for the analogy task *without* requiring 3-way multiplicative connections, with the intent to scale to bigger images and learn more subtle relationships involving articulated pose, multiple attributes and out-of-plane rotation.

Our work is related to several previous works on disentangling factors of variation, for which a common application is analogy-making. As an early example, bilinear models [27] were proposed to separate style and content factors of variation in face images and speech signals. Tang et al. [26] developed the tensor analyzer which uses a factor loading tensor to model the interaction among latent factor groups, and was applied to face modeling. Several variants of higher-order Boltzmann machine were developed to tackle the disentangling problem, featuring multiple groups of hidden units, with each group corresponding to a single factor [23, 7]. Disentangling was also considered in the discriminative case in the Contractive Discriminative Analysis model [24]. Our work differs from these in that we train a deep end-to-end network for generating images by analogy.

Recently several methods were proposed to generate high-quality images using deep networks. Dosovitskiy et al. [10] used a CNN to generate chair images with controllable variation in appear-

ance, shape and 3D pose. Contemporary to our work, Kulkarni et al. [17] proposed the Deep Convolutional Inverse Graphics Network, which is a form of variational autoencoder (VAE) [15] in which the encoder disentangles factors of variation. Other works have considered a semi-supervised extension of the VAE [16] incorporating class labels associated to a subset of the training images, which can control the label units to perform some visual analogies. Cohen and Welling [6] developed a generative model of commutative Lie groups (e.g. image rotation, translation) that produced invariant and disentangled representations. In [5], this work is extended to model the non-commutative 3D rotation group SO(3). Zhu et al. [30] developed the multi-view perceptron for modeling face identity and viewpoint, and generated high quality faces subject to view changes. Cheung et al. [4] also use a convolutional encoder-decoder model, and develop a regularizer to disentangle latent factors of variation from a discriminative target.

Analogies have been well-studied in the NLP community; Turney [28] used analogies from SAT tests to evaluate the performance of text analogy detection methods. In the visual domain, Hwang et al. [13] developed an analogy-preserving visual-semantic embedding model that could both detect analogies and as a regularizer improve visual recognition performance. Our work is related to these, but we focus mainly on generating images to complete analogies rather than detecting analogies.

## 3 Method

Suppose that $\mathcal{A}$ is the set of valid analogy tuples in the training set. For example, $(a, b, c, d) \in \mathcal{A}$ implies the statement "$a$ is to $b$ as $c$ is to $d$". Let the input image space for images $a, b, c, d$ be $\mathbb{R}^D$, and the embedding space be $\mathbb{R}^K$ (typically $K < D$). Denote the encoder as $f : \mathbb{R}^D \to \mathbb{R}^K$ and the decoder as $g : \mathbb{R}^K \to \mathbb{R}^D$. Figure 2 illustrates our architectures for visual analogy making.

### 3.1 Making analogies by vector addition

Neural word representations (e.g., [21, 22]) have been shown to be capable of analogy-making by addition and subtraction of word embeddings. Analogy making capability appears to be an emergent property of these embeddings, but for images we propose to directly train on the objective of analogy completion. Concretely, we propose the following objective for vector-addition-based analogies:

$$\mathcal{L}_{add} = \sum_{a,b,c,d \in \mathcal{A}} ||d - g(f(b) - f(a) + f(c))||_2^2 \tag{1}$$

This objective has the advantage of being very simple to implement and train. In addition, with a modest number of labeled relations, a large number of training analogies can be mined.

### 3.2 Making analogy transformations dependent on the query context

In some cases, a purely additive model of applying transformations may not be ideal. For example, in the case of rotation, the manifold of a rotated object is *circular*, and after enough rotation has been applied, one returns to the original point. In the vector-addition model, we can add the same rotation vector $f(b) - f(a)$ multiple times to a query $f(c)$, but we will never return to the original point (except when $f(b) = f(a)$). The decoder $g$ could (in principle) solve this problem by learning to perform a "modulus" operation, but this would make the training significantly more difficult. Instead, we propose to parametrize the transformation increment to $f(c)$ as a function of both $f(b) - f(a)$ and $f(c)$ itself. In this way, analogies can be applied in a context-dependent way.

We present two variants of our training objective to solve this problem. The first, which we will call $\mathcal{L}_{mul}$, uses multiplicative interactions between $f(b) - f(a)$ and $f(c)$ to generate the increment. The second, which we call $\mathcal{L}_{deep}$, uses multiple fully connected layers to form a multi-layer perceptron (MLP) *without* using multiplicative interactions:

$$\mathcal{L}_{mul} = \sum_{a,b,c,d \in \mathcal{A}} ||d - g(f(c) + W \times_1 [f(b) - f(a)] \times_2 f(c))||_2^2 \tag{2}$$

$$\mathcal{L}_{deep} = \sum_{a,b,c,d \in \mathcal{A}} ||d - g(f(c) + h([f(b) - f(a); f(c)]))||_2^2. \tag{3}$$

For $\mathcal{L}_{mul}$, $W \in \mathbb{R}^{K \times K \times K}$ is a 3-way tensor.[2] In practice, to reduce the number of weights we used a factorized tensor parametrized as $W_{ijl} = \sum_f W_{if}^{(1)} W_{jf}^{(2)} W_{lf}^{(3)}$. Multiplicative interactions

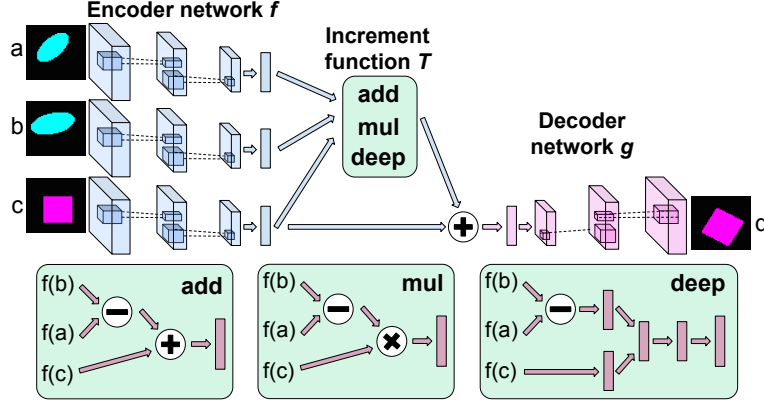

Figure 2: Illustration of the network structure for analogy making. The top portion shows the encoder, transformation module, and decoder. The bottom portion illustrates the transformations used for $\mathcal{L}_{add}$, $\mathcal{L}_{mul}$ and $\mathcal{L}_{deep}$. The $\otimes$ icon in $\mathcal{L}_{mul}$ indicates a tensor product. We share weights with all three encoder networks shown on the top left.

were similarly used in bilinear models [27], disentangling Boltzmann Machines [23] and Tensor Analyzers [26]. Note that our multiplicative interaction in $\mathcal{L}_{mul}$ is different from [19] in that we use the difference between two encoding vectors (i.e., $f(b) - f(a)$) to infer about the transformation (or relation), rather than using a higher-order interaction (e.g., tensor product) for this inference.

For $\mathcal{L}_{deep}$, $h : \mathbb{R}^{2K} \rightarrow \mathbb{R}^K$ is an MLP (deep network without 3-way multiplicative interactions) and $[f(b) - f(a); f(c)]$ denotes concatenation of the transformation vector with the query embedding.

Optimizing the above objectives teaches the model to predict analogy completions in image space, but in order to traverse image manifolds (e.g. for repeated analogies) as in Algorithm 1, we also want accurate analogy completions in the embedding

**Algorithm 1:** Manifold traversal by analogy, with transformation function $T$ (Eq. 5).

Given images $a, b, c$, and $N$ (# steps)
$z \leftarrow f(c)$
**for** $i = 1$ *to* $N$ **do**
$\quad z \leftarrow z + T(f(a), f(b), z)$
$\quad x_i \leftarrow g(z)$
return generated images $x_i$ $(i = 1, ..., N)$

space. To encourage this property, we introduce a regularizer to make the predicted transformation increment $T(f(a), f(b), f(c))$ match the difference of encoder embeddings $f(d) - f(c)$:

$$R = \sum_{a,b,c,d \in \mathcal{A}} ||f(d) - f(c) - T(f(a), f(b), f(c))||_2^2 \text{, where} \tag{4}$$

$$T(x, y, z) = \begin{cases} y - x & \text{when using } \mathcal{L}_{add} \\ W \times_1 [y - x] \times_2 z & \text{when using } \mathcal{L}_{mul} \\ MLP([y - x; z]) & \text{when using } \mathcal{L}_{deep} \end{cases} \tag{5}$$

The overall training objective is a weighted combination of analogy prediction and the above regularizer, e.g. $\mathcal{L}_{deep} + \alpha R$. We set $\alpha = 0.01$ by cross validation on the shapes data and found it worked well for all models on sprites and 3D cars as well. All parameters were trained with backpropagation using stochastic gradient descent (SGD).

### 3.3 Analogy-making with a disentangled feature representation

Visual analogies change some aspects of a query image, and leave others unchanged; for example, changing the viewpoint but preserving the shape and texture of an object. To exploit this fact, we incorporate disentangling into our analogy prediction model. A disentangled representation is simply a concatenation of coordinates along each underlying factor of variation. If one can reliably infer these disentangled coordinates, a subset of analogies can be solved simply by swapping sets of coordinates among a reference and query embedding, and projecting back into the image space. However, in general, disentangling alone cannot solve analogies that require traversing the manifold structure of a given factor, and by itself does not capture image relationships.

In this section we show how to incorporate disentangled features into our analogy model. The disentangling component makes each group of embedding features encode its respective factor of variation and be invariant to the others. The analogy component enables the model to traverse the manifold of a given factor or subset of factors.

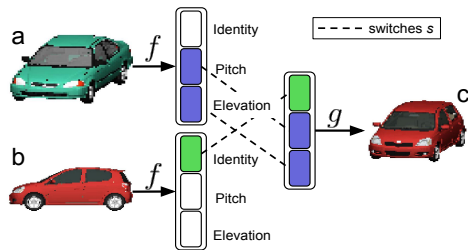

**Algorithm 2:** Disentangling training update. The switches $s$ determine which units from $f(a)$ and $f(b)$ are used to reconstruct image $c$.

Given input images $a, b$ and target $c$
Given switches $s \in \{0,1\}^K$
$z \leftarrow s \cdot f(a) + (1-s) \cdot f(b)$
$\Delta\theta \propto \partial/\partial\theta \left( ||g(z) - c||_2^2 \right)$

Figure 3: The encoder $f$ learns a disentangled representation, in this case for pitch, elevation and identity of 3D car models. In the example above, switches $s$ would be a block $[\mathbf{0}; \mathbf{1}; \mathbf{1}]$ vector.

For learning a disentangled representation, we require three-image tuples: a pair from which to extract hidden units, and a third to act as a target for prediction. As shown in Figure 3, We use a vector of switch units $s$ that decides which elements from $f(a)$ and which from $f(b)$ will be used to form the hidden representation $z \in \mathbb{R}^K$. Typically $s$ will have a block structure according to the groups of units associated to each factor of variation. Once $z$ has been extracted, it is projected back into the image space via the decoder $g(z)$.

The key to learning disentangled features is that images $a, b, c$ should be distinct, so that there is no path from any image to itself. This way, the reconstruction target forces the network to separate the visual *concepts* shared by $(a, c)$ and $(b, c)$, respectively, rather than learning the identity mapping. Concretely, the disentangling objective can be written as:

$$\mathcal{L}_{dis} = \sum_{a,b,c,s \in \mathcal{D}} ||c - g(s \cdot f(a) + (1-s) \cdot f(b))||_2^2 \tag{6}$$

Note that unlike analogy training, disentangling only requires a dataset $\mathcal{D}$ of 3-tuple of images $a, b, c$ along with a switch unit vector $s$. Intuitively, $s$ describes the sense in which $a, b$ and $c$ are related. Algorithm 2 describes the learning update we used to learn a disentangled representation.

## 4 Experiments

We evaluated our methods using three datasets. The first is a set of 2D colored shapes, which is a simple yet nontrivial benchmark for visual analogies. The second is a set of 2D sprites from the open-source video game project called Liberated Pixel Cup [1], which we chose in order to get controlled variation in a large number of character attributes and animations. The third is a set of 3D car model renderings [11], which allowed us to train a model to perform out-of-plane rotation. We used Caffe [14] to train our encoder and decoder networks, with a custom Matlab wrapper implementing our analogy sampling and training objectives. Many additional qualitative results of images generated by our model are presented in the supplementary material.

### 4.1 Transforming shapes: comparison of analogy models

The shapes dataset was used to benchmark performance on rotation, scaling and translation analogies. Specifically, we generated $48 \times 48$ images scaled to $[0, 1]$ with four shapes, eight colors, four scales, five row and column positions, and 24 rotation angles.

We compare the performance of our models trained with $\mathcal{L}_{add}$, $\mathcal{L}_{mul}$ and $\mathcal{L}_{deep}$ objectives, respectively. We did not perform disentangling training in this experiment. The encoder $f$ consisted of 4096-1024-512-dimensional fully connected layers, with rectified linear nonlinearities (relu) for intermediate layers. The final embedding layer did not use any nonlinearity. The decoder $g$ architecture mirrors the encoder, but did not share weights. We trained for 200k steps with mini-batch size 25 (i.e. 25 analogy 4-tuples per mini-batch). We used SGD with momentum 0.9, base learning rate 0.001 and decayed the learning rate by factor 0.1 every 100k steps.

| Model | Rotation steps | | | | Scaling steps | | | | Translation steps | | | |
|---|---|---|---|---|---|---|---|---|---|---|---|---|
| | 1 | 2 | 3 | 4 | 1 | 2 | 3 | 4 | 1 | 2 | 3 | 4 |
| $\mathcal{L}_{add}$ | 8.39 | 11.0 | 15.1 | 21.5 | 5.57 | 6.09 | 7.22 | 14.6 | 5.44 | 5.66 | 6.25 | 7.45 |
| $\mathcal{L}_{mul}$ | 8.04 | 11.2 | 13.5 | 14.2 | 4.36 | 4.70 | 5.78 | 14.8 | 4.24 | 4.45 | 5.24 | 6.90 |
| $\mathcal{L}_{deep}$ | **1.98** | **2.19** | **2.45** | **2.87** | **3.97** | **3.94** | **4.37** | 11.9 | **3.84** | **3.81** | **3.96** | **4.61** |

Table 1: Comparison of squared pixel prediction error of $\mathcal{L}_{add}$, $\mathcal{L}_{mul}$ and $\mathcal{L}_{deep}$ on shape analogies.

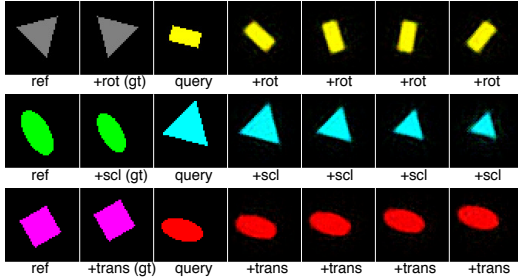

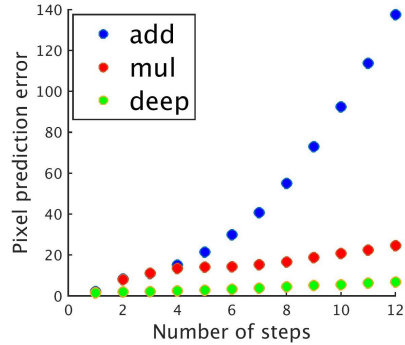

Figure 4: Analogy predictions made by $\mathcal{L}_{deep}$ for rotation, scaling and translation, respectively by row. $\mathcal{L}_{add}$ and $\mathcal{L}_{mul}$ perform as well for scaling and transformation, but fail for rotation.

Figure 5: Mean-squared prediction error on repeated application of rotation analogies.

Figure 4 shows repeated predictions from $\mathcal{L}_{deep}$ on rotation, scaling and translation test set analogies, showing that our model has learned to traverse these manifolds. Table 1 shows that $\mathcal{L}_{add}$ and $\mathcal{L}_{mul}$ perform similarly for scaling and translation, but only $\mathcal{L}_{deep}$ can perform accurate rotation analogies. Further extrapolation results with repeated rotations are shown in Figure 5. Though both $\mathcal{L}_{mul}$ and $\mathcal{L}_{deep}$ are *in principle* capable of learning the circular pose manifold, we suspect that $\mathcal{L}_{deep}$ has much better performance due to the difficulty of training multiplicative models such as $\mathcal{L}_{mul}$.

## 4.2 Generating 2D video game sprites

Game developers often use what are known as "sprites" to portray characters and objects in 2D video games (more commonly on older systems, but still seen on phones and indie games). This entails significant human effort to draw each frame of each common animation for each character.[3] In this section we show how animations can be transferred to new characters by analogy.

Our dataset consists of $60 \times 60$ color images of sprites scaled to $[0, 1]$, with 7 attributes: body type, sex, hair type, armor type, arm type, greaves type, and weapon type, with 672 total unique characters. For each character, there are 5 animations each from 4 viewpoints: spellcast, thrust, walk, slash and shoot. Each animation has between 6 and 13 frames. We split the data by characters: 500 training, 72 validation and 100 for testing.

We conducted experiments using the $\mathcal{L}_{add}$ and $\mathcal{L}_{deep}$ variants of our objective, with and without disentangled features. We also experimented with a disentangled feature version in which the identity units are taken to be the 22-dimensional character attribute vector, from which the pose is disentangled. In this case, the encoder for identity units acts as multiple softmax classifiers, one for each attribute, hence we refer to this objective in experiments as $\mathcal{L}_{dis+cls}$.

The encoder network consisted of two layers of $5 \times 5$ convolution with stride 2 and relu, followed by two fully-connected and relu layers, followed by a projection onto the 1024-dimensional embedding. The decoder mirrors the encoder. To increase the spatial dimension we use simple upsampling in which we copy each input cell value to the upper-left corner of its corresponding $2 \times 2$ output.

For $\mathcal{L}_{dis}$, we used 512 units for identity and 512 for pose. For $\mathcal{L}_{dis+cls}$, we used 22 categorical units for identity, which is the attribute vector, and the remaining 490 for pose. During training for $\mathcal{L}_{dis+cls}$, we did not backpropagate reconstruction error through the identity units; we only used the attribute classification objective for those units. When $\mathcal{L}_{deep}$ is used, the internal layers of the transformation function $T$ (see Figure 2) had dimension 300, and were each followed by relu. We trained the models using SGD with momentum 0.9 and learning rate 0.00001 decayed by factor 0.1 every 100k steps. Training was conducted for 200k steps with mini-batch size 25.

Figure 6 demonstrates the task of animation transfer, with predictions from a model trained on $\mathcal{L}_{add}$. Table 2 provides a quantitative comparison of $\mathcal{L}_{add}$, $\mathcal{L}_{dis}$ and $\mathcal{L}_{dis+cls}$. We found that the disentangling and additive analogy models perform similarly, and that using attributes for disentangled identity features provides a further gain. We conjecture that $\mathcal{L}_{dis+cls}$ wins because changes in certain aspects of appearance, such as arm color, have a very small effect in pixel space yielding a weak signal for pixel prediction, but still provides a strong signal to an attribute classifier.

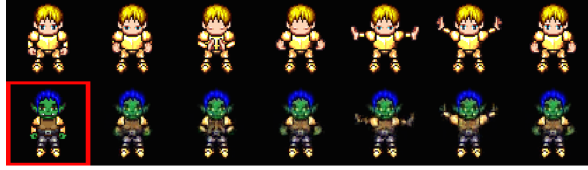

Figure 6: Transferring animations. The top row shows the reference, and the bottom row shows the transferred animation, where the first frame (in red) is the starting frame of a test set character.

| Model | spellcast | thrust | walk | slash | shoot | average |
|---|---|---|---|---|---|---|
| $\mathcal{L}_{add}$ | 41.0 | 53.8 | 55.7 | 52.1 | 77.6 | 56.0 |
| $\mathcal{L}_{dis}$ | 40.8 | 55.8 | 52.6 | 53.5 | 79.8 | 56.5 |
| $\mathcal{L}_{dis+cls}$ | 13.3 | 24.6 | 17.2 | 18.9 | 40.8 | 23.0 |

Table 2: Mean-squared pixel error on test analogies, by animation.

From a practical perspective, the ability to transfer poses accurately to unseen characters could help decrease manual labor of drawing (at least of drawing the assets comprising each character in each animation frame). However, training this model required that each transferred animation already has hundreds of examples. Ideally, the model could be shown a small number of examples for a new animation, and transfer it to the existing character database. We call this setting "few-shot" analogy-making because only a small number of the target animations are provided.

|  | Num. of few-shot examples | | | |
|---|---|---|---|---|
| Model | 6 | 12 | 24 | 48 |
| $\mathcal{L}_{add}$ | 42.8 | 42.7 | 42.3 | 41.0 |
| $\mathcal{L}_{dis}$ | 19.3 | 18.9 | 17.4 | 16.3 |
| $\mathcal{L}_{dis+cls}$ | 15.0 | 12.0 | 11.3 | 10.4 |

Table 3: Mean-squared pixel-prediction error for few-shot analogy transfer of the "spellcast" animation from each of 4 viewpoints. $\mathcal{L}_{dis}$ outperforms $\mathcal{L}_{add}$, and $\mathcal{L}_{dis+cls}$ performs the best even with only 6 examples.

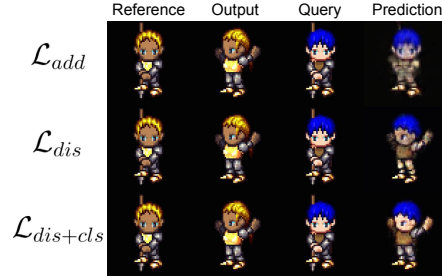

Figure 7: Few shot prediction with 48 examples.

Table 3 provides a quantitative comparison and figure 7 provides a qualitative comparison of our proposed models in this task. We find that $\mathcal{L}_{dis+cls}$ provides the best performance by a wide margin. Unlike in Table 2, $\mathcal{L}_{dis}$ outperforms $\mathcal{L}_{add}$, suggesting that disentangling may allow new animations to be learned in a more data-efficient manner. However, $\mathcal{L}_{dis}$ has an advantage in that it can average the identity features of multiple views of a query character, which $\mathcal{L}_{add}$ cannot do.

The previous analogies only required us to combine disentangled features from two characters, e.g. the identity from one and the pose from another, and so disentangling was sufficient. However, our analogy method enables us to perform more challenging analogies by learning the manifold of character animations, defined by the sequence of frames in each animation. Adjacent frames are thus neighbors on the manifold and each animation sequence can be viewed as a fiber in this manifold.

We trained a model by forming analogy tuples across animations as depicted in Fig. 8, using disentangled identity and pose features. Pose transformations were modeled by deep additive interactions, and we used $\mathcal{L}_{dis+cls}$ to disentangle pose from identity units. Figure 9 shows the result of several analogies and their extrapolations, including character rotation for which we created animations.

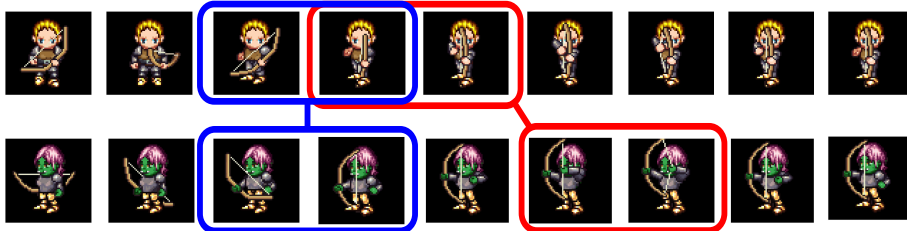

Figure 8: A cartoon visualization of the "shoot" animation manifold for two different characters in different viewpoints. The model can learn the structure of the animation manifold by forming analogy tuples during training; example tuples are circled in red and blue above.

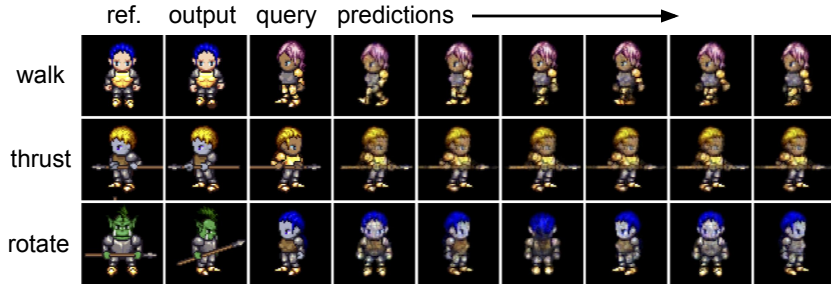

Figure 9: Extrapolating by analogy. The model sees the reference / output pair and repeatedly applies the inferred transformation to the query. This inference requires learning the manifold of animation poses, and cannot be done by simply combining and decoding disentangled features.

## 4.3 3D car analogies

In this section we apply our model to analogy-making on 3D car renderings subject to changes in appearance and rotation angle. Unlike in the case of shapes, this requires the ability of the model to perform out-of-plane rotation, and the depicted objects are more complex.

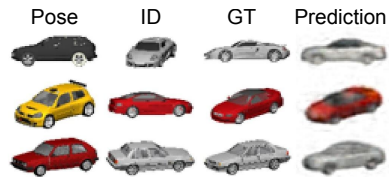

| Features | Pose AUC | ID AUC |
|---|---|---|
| Pose units | **95.6** | 85.2 |
| ID units | 50.1 | **98.5** |
| Combined | 94.6 | 98.4 |

Table 4: Measuring the disentangling performance on 3D cars. Pose AUC refers to area under the ROC curve for same-or-different pose verification, and ID AUC for same-or-different car verification on pairs of test set images.

Figure 10: 3D car analogies. The column "GT" denotes ground truth.

We use the car CAD models from [11]. For each of the 199 car models, we generated $64 \times 64$ color renderings from 24 rotation angles each offset by 15 degrees. We split the models into 100 training, 49 validation and 50 testing. The same convolutional network architecture was used as in the sprites experiments, and we used 512 units for identity and 128 for pose.

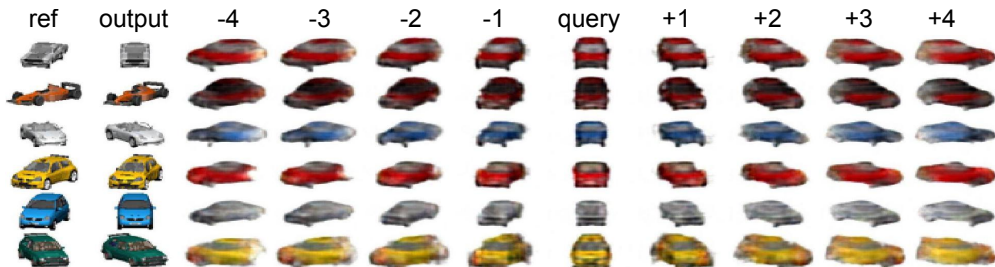

Figure 11: Repeated rotation analogies in forward and reverse directions, starting from frontal pose.

Figure 10 shows test set predictions of our model trained on $\mathcal{L}_{dis}$, where images in the fourth column combine pose units from the first column and identity units from the second. Table 4 shows that the learned features are in fact disentangled, and discriminative for identity and pose matching despite not being discriminatively trained. Figure 11 shows repeated rotation analogies on test set cars using a model trained on $\mathcal{L}_{deep}$, demonstrating that our model can perform out-of-plane rotation. This type of extrapolation is difficult because the query image shows a different car from a different starting pose. We expect that a recurrent architecture can further improve the results, as shown in [29].

## 5 Conclusions

We studied the problem of visual analogy making using deep neural networks, and proposed several new models. Our experiments showed that our proposed models are very general and can learn to make analogies based on appearance, rotation, 3D pose, and various object attributes. We provide connection between analogy making and disentangling factors of variation, and showed that our proposed analogy representations can overcome certain limitations of disentangled representations.

**Acknowledgements** This work was supported in part by NSF GRFP grant DGE-1256260, ONR grant N00014-13-1-0762, NSF CAREER grant IIS-1453651, and NSF grant CMMI-1266184. We thank NVIDIA for donating a Tesla K40 GPU.

## Footnotes

[1]See [2] for a deeper philosophical discussion of analogical reasoning.

[2]For a tensor $W \in \mathbb{R}^{K \times K \times K}$ and vectors $v, w \in \mathbb{R}^K$, we define the tensor multiplication $W \times_1 v \times_2 w \in \mathbb{R}^K$ as $(W \times_1 v \times_2 w)_l = \sum_{i=1}^K \sum_{j=1}^K W_{ijl} v_i w_j, \forall l \in \{1, ..., K\}$.

[3]In some cases the work may be decreased by projecting 3D models to 2D or by other heuristics, but in general the work scales with the number of animations and characters.

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
