[Supplementary Material · supplement.pdf]

# Supplementary Material:
# Deep Visual Analogy-Making

**Scott Reed     Yi Zhang     Yuting Zhang     Honglak Lee**
University of Michigan, Ann Arbor, MI 48109, USA
{reedscot,yeezhang,yutingzh,honglak}@umich.edu

This document contains qualitative results that could not fit into the main text.

## 1   Trajectories of multiple shape analogies

Our model can apply analogies in both the forward and reversed direction. In Figure 1, the first two columns indicate the operation, i.e., the relationship to be applied to the query. The first row demonstrates several steps of clockwise rotation, followed by counter-clockwise, returning to the initial orientation. The second and third rows show that our model can perform the same feat for scaling and translation. Although we only trained for 1-step analogies, the model is able to stay on the manifold even after repeated transformations.

Figure 1: Repeated application of analogies from the example pair (first two columns), in both forward and reverse mode, using a model trained with $\mathcal{L}_{deep}$.

We can also interleave different kinds of transformations by supplying multiple pairs of transformed images. Figure 2 shows interleaving of rotation, translation and scaling in the same sequence.

Figure 2: Repeated application of multiple different analogies from two different example pairs (first four columns) using a model trained with $\mathcal{L}_{deep}$.

Figure 1 and Figure 2 are also available in video format: shape-*.avi

## 2   Comparing $\mathcal{L}_{add}$, $\mathcal{L}_{mul}$, and $\mathcal{L}_{deep}$ for shape analogies

Following the protocol in the previous section, we apply the models trained with $\mathcal{L}_{add}$, $\mathcal{L}_{mul}$, and $\mathcal{L}_{deep}$ for multi-step shape analogies. As it is shown in Figure 3, the model learned with $\mathcal{L}_{add}$ and $\mathcal{L}_{mul}$ cannot do very well for even 1-step rotation, while the model learned with $\mathcal{L}_{deep}$ can support manifold traversal. Scaling and translation are relatively simple for $\mathcal{L}_{add}$ and $\mathcal{L}$, but the qualitative degradation due to multi-step analogies is still noticeably more significant than that of $\mathcal{L}_{deep}$.

Figure 3: Repeated application of analogies from the example pair (first two columns), in both forward and reverse mode, using three models trained respectively with $\mathcal{L}_{add}, \mathcal{L}_{mul}, \mathcal{L}_{deep}$.

# 3 Fine-grained control over sprite attributes

In Figure 4, we show how disentangling and attribute classification objectives can help with fine-grained control on the discrete-valued attributes of generated sprites.

Figure 4: Using $\mathcal{L}_{dis+cls}$, our model can generate sprites with fine-grained control over character attributes. The above images were generated by using $f$ to encode the leftmost source image for each attribute, and then changing the identity units and re-rendering.

# 4 Animation transfer using disentangled features

When we have a model trained by $\mathcal{L}_{dis}$ or $\mathcal{L}_{dis+cls}$, we can extract disentangled identity and pose features for sprites. Performing pose transfer simply requires taking the pose of a reference image and the identity of a query image, and using the decoder $g$ to project their combination back into the image space. In Figures 5, 6 and 7, we show several consecutive frames of pose transfer, which we call animation transfer since we can follow the entire trajectory of an animation.

Figure 5: Shooting a bow.

Figure 6: Walking.

Figure 7: Casting a spell.

# 5  Animation transfer videos

We have also included several video clips of reference animations and transferred generated animations. They are named as follows:

```
videos/walking_ref.avi : reference animation of walking.
videos/walking_gen.avi : generated animation with the ID units extracted
                         from output frame 1, and the poses of each
                         subsequent frame taken from the reference.
videos/shoot_ref.avi   : reference animation of shooting a bow.
videos/shoot_gen.avi   : generated animation
videos/thrust_ref.avi  : reference animation of thrusting a spear.
videos/thrust_gen.avi  : generated animation
```

# 6  Sprite animation analogies with extrapolation

Below in Figure 8, we show cross-identity animation extrapolations for each of the five animations, plus rotation. The analogy model has learned the structure of the animation manifolds across variations in viewpoint and character attributes, and is able to advance forward or backward in time based on the example image pair.

Figure 8: Animation analogies and extrapolation for all character animations plus rotation. The example pair (first two columns) and query image (third column) both come from the test set of characters. $\mathcal{L}_{deep}$ was used for analogy training of pose units, jointly with $\mathcal{L}_{dis+cls}$ to learn a disentangled representation.