[Reviews · NeurIPS 2015]

Submitted by Assigned_Reviewer_1

This is a great paper; clear accept.

Quality: great

Clarity: The entire paper is well written. Equations are given only when needed. Fig 2 is excellent.

Originality: Analogy making has been studied, as have decoding architectures for images. Their unique contributions are twofold: first, they show both analogy making and encoding/decoding working together, end to end. Next, note that Mikolov (NIPS 2013) showed the intriguing result that using some methods to produce embeddings (in their case, skip grams) results in a space in which analogies are approximately linear. This has lead researchers on a quest for other methods of embedding different types of data in spaces where analogies are linear. Critically, this paper shows that even in cases where training toward a linear embedding doesn't work so well (top row of Table 1), it may be possible or easy to train toward embeddings that can be reasoned about through a non-linear operation, here a deep neural net.

Significance: Likely to be received positively by the field and promote further work. The latter result above is important for future work in analogies.
Summary: Great paper showing analogical reasoning for images using encoder/decoder pairs. Linear (ala Mikolov 2013) and non-linear neural net analogy models are tested.

Submitted by Assigned_Reviewer_2

This paper discusses the ambitious problem of creating generative models that can transform images according to analogies given by example pairs of images.

Based on the success of word2vec, this paper takes a very simple approach of representing analogies as additions in a highly abstract feature space followed by a decoder method. However images do not have the same

contextual consistency as text, so the paper proposes a very simple approach that regresses towards the decoded transformed image: a squared L2 loss is used the compare with the the transformed image, where the transformation is represented by the difference of the features of the template images from which the analogy is derived from. The analogy direction is exploited and compared in three different ways: as simple additions or multiplications over the feature vectors, or by using a suitably constructed deep combiner

network. The paper presents compelling evidence on two simple well controlled benchmarks that deep combiners utilizing switches to disentangle features works best.

The paper is written well and clearly. Although it is relatively straightforward, it makes a fundamental contribution to the important emerging domain of visual analogy making by analyzing the most direct ways of attack.
Summary: This paper is well written and although the technical content is

mathematically simple, it provides a very elegant method to the fundamental and important problem of visual analogy making by end-to-end training a deep network that directly addresses the task.

Submitted by Assigned_Reviewer_3

This paper proposes a method to find analogical images using given query images. Specifically, given four images a, b, c, and d, we would like to find a transformed version of c such that it is similar to the transformation from a to b when applied to d. This is an interesting problem and should be of interest to image recognition community in general because if images could be represented in a space (vectors in the proposed method), then such translation invariant representations could simplify the task of recognizing images thereby improving recognition accuracies.

Although using analogical methods for image recognition is a relatively new topic in the visual domain, it has a well established history in NLP. For example, SAT word analogy questions have been used to evaluate the performance of analogy detection methods. See @article{Turney_CL,

Author = {P.D. Turney},

Journal = {Computational Linguistics},

Number = {3},

Pages = {379--416},

Title = {Similarity of semantic relations},

Volume = {32},

Year = {2006}} for an overview. Both supervised and unsupervised methods have been proposed for this purpose. Some background description of such prior work on analogy detection will give a good context to the current paper.

Moreover, some work has been done in the visual domain studying embeddings that preserve analogical properties in images. @inproceedings{Hwang:ICML:2013,

Author = {Sung Ju Hwang and Kristen Grauman and Fei Sha},

Booktitle = {ICML'13},

Title = {Analogy-preserving Semantic Embedding for Visual Object Categorization},

Year = {2013}}

Sec 3.1 argues that sampling from co-occurring contexts results in word embeddings that demonstrate analogical properties when the vector difference is considered. However, this claim is not clear to me. Why would predicting word co-occurrence result in word representations that demonstrate analogies? It is true empirically that this is the case but why predicting co-occurrences should help?

In the proposed method this is by design. In other words, the objective functions are designed in such a way to minimize the loss between the target image d and its construction via a transform on a, b, and c. There is a recent work that learns semantic representations for words by predicting analogies directly in similar spirit to the proposed method. Embedding semantic relations into Word Representations IJCAI 2015 arxiv paper.

In Algo 1, once you create N candidate images how would you select the best one for d? Do you evaluate the objective function for each candidate and select the best one? A related question is how to decide N? Does it depend on the difference between a and b?

One weakness of the evaluation section is that all evaluations are limited to different baselines/settings within the proposed method. It would have been good if there was some comparison against prior work.
Summary: This paper proposes a method for finding analogical images for a query image.

Submitted by Assigned_Reviewer_4

4.1 How did the train, valid/test were divided? To show that the model learns the transformations, some tuples (a, b, c) should never been seen in training set, otherwise one can claim that the model only has memorized

the training set.

L 309: What kind of up sampling? Please elaborate. -for the decoder, was any sort of deconvolution was used, or it was normal convolution? -It would be very interesting to see how a model like VAE do on the tasks in the paper. One could train a VAE just on the single images and limit the latent space to the degrees of freedom in the dataset. Then it should be easy to control the latent space and generate desired outputs. -[1] might not be directly related, but similar to word2vec they also learn multi-modal embeddings that results in very interested results when algebraically combined.

1. Unifying Visual-Semantic Embeddings with Multimodal Neural Language Models

Ryan Kiros, Ruslan Salakhutdinov, Richard Zemel. TACL, 2015
Summary: The authors propose a simple but elegant architecture to learn analogies on images and be able to generate new image given a triplet input. They have two family of models, one that learn by looking at two pair of images (a, b), (c, d) and learn a->b transformation and apply it to c to generate d. And the other model that only looks at (a, b, c) and a switch vector s which encodes the underlying factors of variations in a and b. This is a very interesting idea and I enjoyed its simplicity, and the experiments shows promising results. The main shortcoming is that they do not empirically compare their results with any prior work that they cite.

Author Feedback
Author rebuttal: We thank all reviewers very much for your thoughtful comments. We will revise the paper reflecting all your comments.

R2:
Yes, in the multiplicative version f(c) can be viewed as gating W, and the resulting query-specific matrix is multiplied by f(b) - f(a).

We did not yet try using a cross-entropy loss, but it is straightforward to change the loss, and this could be useful in future work.

Thank you for your comments regarding 2D projections of the embeddings. It indeed seems to reveal an interesting structure in our preliminary experiments. For example in the rotating shapes, we observe a "ring" structure. In other words, the trajectory of the traversal in the projected embedding looks like a ring where 360 degree rotation maps to approximately the starting point (corresponding to "no rotation") in the embedding space.

Regarding the alternative discriminative tasks, we suspect that the learned embeddings + linear classifier should do a good job, although we have not empirically validated this. That being said, using task#1 and task#2 directly could be effective regularizers in addition to task#3.

R3:
Thank you for pointing us to those papers, we will incorporate them into the related work.

Regarding the mention of co-occurrence statistics yielding analogy properties, the idea is that if you view word embeddings of an analogy tuple (a,b,c,d) as particles in a dynamical system during training, many word embedding algorithms exert an attractive force on co-occurring words and repulsive force on non-co-occurring words. We conjecture that the dynamics are such that the particles (a,b,c,d) tend to form an approximate parallelogram (yielding vector analogies), although we agree that this is perhaps too speculative. We are happy to remove this from the final version.

Regarding how to choose the number of steps N for manifold traversal: this depends on the difficulty of the problem. For 2D shapes we found the model can, e.g. perform full 360 degree rotation easily, but for more challenging datasets like 3D cars the generated image quality decays before 180 degrees. Note that during training, N=1 by definition, since we only train to perform a single-step analogy completion. The ability to extrapolate along the manifold differentiates the "add", "mul" and "deep" versions of our model; namely that "deep" achieves the best performance on repeated analogies. To increase the viable N for more challenging data, in future work we aim to incorporate explicit recurrent structure.

R4:
Regarding split - yes, the testing tuples (a,b,c) were kept separate from the training tuples. Furthermore, we trained only on 1-step analogies and tested on multi-step analogies.

For the upsampling we followed the chair-generating CNN (Dosovitskiy et al., CVPR 2015), i.e. for a 1x1 patch expanded to a 2x2 patch, we copy its value to the upper-left cell and leave the rest zero. This is of course followed by further convolution and upsampling.

In the decoder convolutions, we first padded the input so that the spatial dimension did not change; i.e. only the upsampling operation changes the spatial dimension, except for a final "valid" convolution after the final upsampling.

We agree that exploring a VAE variation on the model would be interesting and potential future work. We will add [1] to our related work section.

R5:
The disentangling and analogy objectives are combined linearly. In practice we alternately sampled training mini-batches for the disentangling and analogy objective.